- Soil signals of key mechanisms driving greater protection of organic carbon under aspen
- compared to spruce forests in a North American montane ecosystem
- Authors: Lena Wang<sup>1</sup>, Sharon A. Billings<sup>2</sup>, Li Li<sup>3</sup>., Daniel R. Hirmas<sup>4</sup>, Keira Johnson<sup>1</sup>, Devon
- Kerins<sup>3</sup>, Julio Pachon<sup>5</sup>, Curtis Beutler<sup>6</sup>, Karla M. Jarecke<sup>1</sup>, Vaishnavi Varikuti<sup>4</sup>, Micah Unruh<sup>2</sup>,
- Hoori Ajami<sup>7</sup>, Holly Barnard<sup>8</sup>, Alejandro N. Flores<sup>9</sup>, Kenneth Williams<sup>6</sup>, Pamela L. Sullivan<sup>1</sup>

- College of Earth Ocean and Atmospheric Science, Oregon State University. Address: 101 SW
  26th St, Corvallis, OR, 97331, USA
- <sup>2</sup>Department of Ecology and Evolutionary Biology and Kansas Biological Survey & Center for
- Ecological Research, University of Kansas. Address: 2101 Constant Ave., Lawrence, KS, 66047,
- USA
- <sup>3</sup>Department of Civil and Environmental Engineering, Pennsylvania State University,
- Address: Sackett, #212, University Park, PA 16802, USA.
- <sup>4</sup>Department of Plant and Soil Science, Texas Tech University. Address: 2500 Broadway
- Lubbock, Texas 7940 USA.
- <sup>5</sup>Sydney Institute of Agriculture, School of Life and Environmental Sciences, University of
- Sydney, New South Wales, Australia. Address: The University of Sydney, NSW 2006, Australia.
- <sup>6</sup>Lawrence Berkeley National Laboratory, Berkeley, CA USA Rocky Mountain Biological
- Laboratory, Gothic, CO USA. Address: 1 Cyclotron Road, Berkeley, CA 94720, USA.
- <sup>7</sup>Department of Environmental Sciences, University of California Riverside. Address: 2460B
- Geology Building, Riverside, CA 92521USA.
- <sup>8</sup>Department of Geography, Institute of Arctic and Alpine Research, University of Colorado –
- Boulder. Address: Guggenheim 110, 260 UCB Boulder, Colorado 80309-0260, USA.
- <sup>9</sup>Department of Geosciences, Boise State University, Boise Address: 1295 University Drive,
- Boise, ID 83706 USA.
- Correspondence to: Pamela L. Sullivan(Sullipam@oregonstate.edu)

# 29 Highlights:

343536

56

59 60

- 1. Soil organic carbon concentrations are consistently greater under aspen compared to spruce
- 2. Microbial Efficiency Matrix Stabilization model helps explain SOC differences
- 3. Smaller aggregate sizes under aspen further help explain SOC concentrations
- 4. A lower probability of SOC destabilization likely persists under aspen stands

## Abstract

Soil organic carbon (SOC) is often retained more effectively in aspen-dominated forests 38 compared to coniferous forests in North America, yet the reasons why are unclear. A potential 39 driver could be differences in SOC protection mechanisms. Over decades to centuries, chemical 40 (e.g., mineral association) and physical (e.g., aggregation) processes can work to preserve SOC stocks, which can vary across cover types. To investigate this hypothesis, we evaluate controls 41 42 on SOC concentrations in the Coal Creek watershed (CO, USA), a montane ecosystem 43 dominated by quaking aspen and Engelmann spruce and underlain by granite and sandstone. 44 We examined a combination of biological, chemical, physical, and environmental conditions to 45 evaluate potential abiotic and biotic mechanisms of SOC preservation at multiple depths. As 46 expected, we observed greater SOC concentrations under aspen compared to spruce. Growing 47 season soil moisture, temperature, and CO<sub>2</sub> and O<sub>2</sub> varied with slope position and aspect, and 48 thus forest cover type. Dissolved organic carbon (DOC) was lower under aspen compared to 49 spruce. Exo-enzyme data indicate that aspen soil microbes likely access more organically-bound 50 resources; consistent with this, soil organic N exhibited higher  $\delta^{15}$ N values, hinting at a greater degree of organic matter processing. Finally, aspen soils exhibited greater root abundance, and 51 52 aspen mineral soils revealed smaller mean aggregate diameters compared to conifer sites. Our 53 data suggest enhanced biotic activities in aspen-dominated forest soils that promote both 54 chemical and physical protection of SOC in aspen-relative to spruce-dominated forests, which 55 may have implications for DOC export.

Keywords (1-7 words): Critical Zone, Ecohydrology, Montane Ecosystems, Soil Organic Carbon,
 Climate Change

### 1 Introduction

The distribution and composition of temperate montane forests are changing (Alexander et al., 1987, Anderegg et al., 2013), driven by increasing air temperature, earlier snowmelt, earlier onset and extent of the growing season (Godsey et al., 2014; Mote et al., 2018; Rhoades et al., 2018), and increasing frequency and intensity of disturbance (e.g., drought, fire, logging, and insect infestation; Canelles et al., 2021). For example, aspen stands have lost substantial live density and basal area to Englemann spruce, sub-alpine fir, and Douglas Fir since 1964 with an

67 increasing rate of decline since 1994 (Alexander, 1987; Coop et al., 2014). Changes in montane 68 forest cover can directly impact soil organic carbon (SOC) stability. Given that SOC influences 69 the availability of nutrients, soil stability, ecosystem water fluxes, and biosphere-atmosphere 70 exchange of greenhouse gases (Jackson et al., 2017), and that global SOC reservoirs represent 71 far more C than plant biomass and the atmosphere (Scharlemann et al., 2014), unraveling 72 drivers of SOC stability remains an important research goal (Billings et al., 2021). Between 73 paired aspen and conifer stands at numerous sites throughout North America, SOC pools differ 74 substantially (review in Langaniere et al., 2017). Studies consistently show that C under conifers 75 is more readily destabilized than under aspen (Woldeselassie et al., 2012; Laganiere et al., 2013; 76 Boča et al., 2020; Román Dobarco et al., 2021). Further, SOC pools in aspen-dominated 77 environments tend to be composed of larger stocks of mineral-associated organic C (MAOC), 78 which is a relatively stable SOC fraction, than those under conifers (66% compared to 48% 79 MAOC to SOC, respectively; Román Dobarco and Van Miegroet, 2014; Román Dobarco et al., 80 2021). Yet, the mechanisms driving such differences in SOC stability under aspen and conifer 81 remain elusive. 82 Examining soil physical attributes and how they can differ with plant cover type may help us 83 84 85 aggregation refers to the clustering or binding of soil particles into larger units. This process is 86

understand differences in MAOC fate in aspen vs. conifer forests. For example, soil aggregation is a key process promoting SOC protection in many soil types (Blanco-Canqui and Lal, 2004). Soil aggregation refers to the clustering or binding of soil particles into larger units. This process is promoted by interactions among colloidal material and binding compounds (microaggregates; Six et al., 2004; Blanco-Canqui and Lal, 2005; Weil and Brady, 2017; Araya and Ghezzehei, 2019) and among particulate organic C (POC; Cotrufo et al., 2019), and clay minerals or clay-sized particles (Six et al., 2000). The collapse and formation of aggregates influence the protection of SOC. For example, the breakdown of macroaggregates into microaggregates often leads to the release of dissolved organic C (DOC) (Cincotta et al., 2019), some of which can undergo microbial uptake and mineralization to CO<sub>2</sub>. In contrast, aggregate formation can limit soil microbial access to SOC on aggregates' interiors, helping to shield it from exo-enzymatic attack (Jastrow 1996; Six et al., 2000; Woolf et al., 2019).

Multiple mechanisms may drive differences in soil aggregation across aspen and conifer soils. 96 First, soils beneath conifers are often more acidic (Poponoe et al., 1992; Buck and St. Clair, 97 2012) and thus may promote a greater abundance of relatively small aggregates, given that 98 increases in soil solution [H<sup>+</sup>] can weaken soil aggregation processes (Stătescu et al., 2013). 99 Second, differences in rooting abundance among aspen and conifers may drive differences in 100 aggregate formation across these cover types. Aspen tends to produce shallow roots that 101 generally extend to ~ 30 cm deep (Sheppard et al., 2006), while conifers tend to develop both 102 lateral and tap roots, the latter of which can extend relatively deep into the soil and bedrock 103 (Mauer et al., 2012). Spruce tends to exhibit lower fine root biomass compared to aspen

104 (Mekontchou et al., 2020). Fine roots may promote aggregate formation through enmeshment 105 processes, while coarse roots may promote aggregate collapse because of roots perforating 106 aggregates (Bronick and Lal, 2005). Differences in soil moisture between aspen and conifers 107 driven by differences in aspect, foliar cover, and transpiration rates (Buck and St. Clair, 2012) 108 may also influence aggregate stability, as rapid changes in soil moisture can cause aggregates to 109 burst while a gradual increase in moisture can stabilize aggregates (Amezketa, 1999). 110 Combined, these concurrent and competing processes may drive differences in soil aggregation 111 between aspen- vs. conifer-dominated soils in ways that are difficult to predict due to complex 112 and non-linear interactions, and require the synthesis of findings across biological and 113 pedological disciplines to understand. 114 Soil moisture and temperature not only influence physical aggregation processes, and thus the 115 protection and preservation of SOC, but also the degree to which microbes transform SOC into 116 CO<sub>2</sub> or alter the transport of organic C pools to depth. Where soil moisture is higher, greater 117 transport of organic C pools into the subsurface may be feasible, potentially increasing the 118 amount of organic matter sorbed to minerals at greater depths (Mikutta et al., 2019). 119 Conversely, DOC leaching may increase, and subsequent DOC export could reduce SOC 120 concentrations (Roulet and Moore, 2006; Monteith et al., 2007). Soil temperature also may 121 drive differences in SOC transformations across aspen and conifer sites, given that aspect exerts 122 strong control on aspen distribution. Soil temperatures tend to be warmer under the sunnier, 123 aspen-dominated stands compared to conifer stands (Buck and St. Clair, 2012). In a 124 temperature-limited montane system, warmer temperatures under aspen stands may increase 125 microbial metabolic activity and turnover, and thus accelerate microbial necromass formation, 126 a process linked to greater stocks of relatively persistent SOC (Liang et al., 2019), perhaps due 127 to necromass-promoting aggregate formation and stabilization (Sae-Tun et al., 2022). Thus, 128 understanding soil water movement and solute transport—traditionally studied by hydrologists 129 and soil biogeochemists—along with biological and soil formation processes, is key to 130 explaining patterns of SOC transformations. 131 Finally, differences in the chemical composition of aspen and conifer biomass and their root 132 exudates may explain differences in MAOC stocks between the two stand types (Boča et al., 133 2020). For example, aspen litter tends to exhibit lower lignin concentrations than coniferous 134 litterfall (Moore et al., 2006). The Microbial Efficiency - Matrix Stabilization (MEMS) framework 135 (Cutrofo et al., 2013) would suggest this more labile plant material may be easier for soil microbes to assimilate and transform into microbial necromass, which can become more 136 137 physically or chemically protected through aggregation and chemical bonding (Kleber et al., 138 2007; von Lützow et al., 2008; Cutrofo et al., 2013) and lead to relatively more persistent stocks 139 of SOC (Liang et al., 2019; Buckridge et al., 2022). Differences in microbial activities between 140 aspen and conifer may further be exacerbated by differences in root exudation between these

141 species. For example, Norway spruce can exhibit lower exudation rates than silver birch (Sadnes 142 et al., 2005), and deciduous trees appear to experience greater exudation rates than pines 143 (Wang et al., 2021). Though many studies explore the biotic, chemical, physical, and hydrologic 144 processes that can influence SOC transformations and preservation, these processes are rarely 145 examined at the same time. Thus, it remains unclear why conifer-dominated forests consistently harbor smaller amounts of SOC, and why aspen-dominated forests exhibit greater 146 147 SOC stabilization. 148 Here, we use a holistic, critical-zone approach —integrating physical, chemical, and biological 149 processes from the vegetation canopy to bedrock (Chorover et al., 2007)—drawing on data 150 from biology, hydrology, pedology, and other disciplines to understand SOC dynamics and 151 drivers. We explore a suite of abiotic and biotic factors as they relate to SOC pool sizes 152 across two forest cover types at Coal Creek, a watershed in central Colorado, USA, dominated 153 by Englemann spruce (Picea engelmanni) on the north-facing hillslopes, and aspen (Populus 154 tremuloides) on the south-facing hillslopes. Coal Creek has experienced relatively high 155 variability in stream water DOC concentrations in recent years (2005-2019; Leonard et al., 156 2022). The mysterious, almost tripling of stream DOC concentrations in some years (2018-2019) 157 may indicate recent shifts in upslope biogeochemical processes such as greater forest stress 158 associated with climate change (Leonard et al., 2022) and subsequent changes in hydrologic 159 flow paths (Zhi et al., 2020; Kerins et al., 2023) that influence C transport from soil profiles to 160 stream water. We test the hypothesis that higher soil organic carbon (SOC) stocks commonly 161 observed under aspen stands—relative to conifer-dominated soils—are driven by enhanced 162 microbial activity in aspen soils. We further hypothesize that aspen-dominated soils contain 163 more stable microaggregates than spruce soils, driven by higher microbial activity and 164 associated increases in necromass production. Differences in rooting strategies and 165 ecohydrologic factors (e.g., evapotranspiration, soil moisture) between aspen and spruce likely 166 exert secondary controls on C stability. Finally, we hypothesize that the proliferation of fine 167 roots in aspen soils is associated with smaller water-stable aggregates, whereas the deeper, 168 coarser roots in spruce soils promote vertical movement of water and dissolved C down 169 through the soil profile, potentially leading to greater DOC export to streams compared to 170 aspen systems.

To test these hypotheses, we quantified multiple metrics describing basic abiotic conditions, SOC pools, soil microbial activities, soil aggregate-size distributions, and rooting distributions on five hillslopes dominated by either spruce or aspen, underlain by two contrasting lithologies and located at two hillslope positions (i.e., backslope and footslope). We aim to clarify some of the mechanisms governing aspen- and conifer-dominated forest soil microbial activity, soil aggregation, and soil moisture dynamics and their impact on SOC protection and potential DOC transport into surface water, illuminating the possible trajectories of SOC and DOC in rapidly changing, montane forest watersheds.

## 2 Study Area

183 Figure 1: A map of the Coal Creek catchment. Colors represent land cover types, where aspen (orange)

are dominantly at lower south-facing slopes while conifer (green) are on both north and south facing

slopes. Shapes represent lithology type where granite sites (blue triangles) are in the western part of the

- catchment and sandstone sites (pink circles) are in the eastern part of the catchment. AS is aspen
- sandstone, and SS is spruce sandstone. ESG and EAG are spruce granite and aspen granite, respectively.
- They are in Elk Creek, a sub-catchment of Coal Creek. While ESG is on a dominantly south facing slope, it
- is north facing within the Elk Creek catchment. SG is also a spruce granite site. Note that all sites reside
- at contrasting hillslope positions: backslope = AS and SG, and footslopes = SS, ESG, and EAG.
- Coal Creek (53 km<sup>2</sup>) is a high-elevation (2715 m), headwater tributary of the Upper Colorado
- River Basin located in the Colorado Rocky Mountains near the town of Crested Butte (Fig. 1).
- Coal Creek is a sub-catchment of the larger East River watershed (300 km<sup>2</sup>) and falls within the
- research domains of the U.S. Department of Energy funded Watershed Function Science Focus
- Area and Rocky Mountain Biological Laboratory (RMBL). The watershed is seasonally snow-
- covered from November through June. The area has a continental, subarctic climate with long,
- cold winters and short, cool summers. The mean annual temperature is 0.9 °C and the mean
- annual precipitation is 670 mm (Carroll et al., 2018), with approximately 60% falling as snow
- between October and May. This area has been warming since the 1980s and the fraction of
- snow has been decreasing roughly at 1% per year (Zhi et al., 2020). Due to these warming
- temperatures, the growing season in Crested Butte appears to be extending (Wadgymar et al.,
- 2018).
- The geology of Coal Creek is underlain by sandstone, siltstone, shale, and coal units from the
- Mesa Verde Formation, variegated claystone and shale from the Wasatch Formation, and some
- intrusive granite diorite, granite, quartz, and monzonite that are Middle Tertiary aged (Gaskill et
- al., 1991). Soils are predominantly mapped as carbonate free Alfisols, Mollisols, and Inceptisols
- (Soil Survey Staff, 2023).
- Spruce, aspens, and alpine meadows can be found in the Coal Creek watershed. North-facing
- slopes are dominated by Engelmann Spruce, while aspen and Engelmann spruce can be found
- on south-facing slopes. We focused on five sites during this study. Three of our sites lie within
- the main drainage of Coal Creek including two spruce sites (spruce sandstone, SS; spruce
- granite, SG) and one aspen (aspen sandstone (AS). The last two sites are located in Elk Creek, a
- sub-catchment of Coal Creek, which includes one spruce site and one aspen site, both underlain
- by granite (Elk spruce granite (ESG) and Elk aspen granite (EAG). While ESG is on a dominantly
- south facing slope, it is north facing within the Elk Creek catchment.

## 217 3 Methods

- To quantify the impact of aspen vs. conifer land cover on soil organic C dynamics at Coal Creek,
- we dug two pits roughly one meter deep at all five sites. The first series of pits were dug in the
- summer 2020 and 2021 (Table 1). The second series of pits were dug in the summer of 2022.
- Aspen was dominant at two sites (AS and EAG), while spruce was dominant at three sites (SS,
- ESG, and SG). Because aspen- and conifer-dominated forests in this region tend to occur on
- hillslopes of contrasting aspects, it was not possible to isolate land cover from aspect effects
- (e.g., temperature, radiation). While the sites were selected based on their land cover, other

key ecosystem features underlying lithology (either granite or sandstone), and hillslope position (either backslopes or footslopes) (Fig. 1) also differed across the sites. We address these site features as potential sources of variation in our response variables in the discussion.

Soil pits were described following Schoeneberger et al. (2012), then each pit face was photographed with a high-resolution, digital single-lens reflex camera (D5600, Nikon, Minato City, Tokyo, Japan) to quantify rooting depth distributions following Billings et al. (2018). Bulk soil samples were collected by depth every 10 cm for the first set of pits (2020-2021), and by horizon for the second set of pits (2022). Samples were then immediately stored in a refrigerator or freezer (DOC, microbial biomass C, exo-enzyme assays, nitrate) until they could be ground, sieved to 2 mm and analyzed. Twice in the summer of 2022 (late June and mid-August), soil was collected from 3 auger sampling locations within ~100 meters of each pit to characterize soil chemistry (i.e., SOC, DOC, pH). Soils were augured at 10 cm intervals to 110 cm (or deepest possible depth), and samples were stored in coolers with ice packs in the field and transported back to the lab and stored at 4 °C (most analyses) or frozen (SOC, DOC).

Table 1. Sampling design and analysis for the soil pits and augers samples.

|                                                    | Call Dita                            | Cail Dita    | Auger                        |
|----------------------------------------------------|--------------------------------------|--------------|------------------------------|
|                                                    | Soil Pits                            | Soil Pits    | Samples                      |
| Timing                                             | 2020-2021<br>(July and<br>September) | 2022<br>July | 2022<br>(June and<br>August) |
| Total Depth (cm)                                   | ~110                                 | ~110         | ~110                         |
| Soil Sampling Intervals                            | 10 cm                                | Horizon      | 10 cm                        |
| Soil moisture and gas sensors                      | 3 depths<br>(15, 45, 110<br>cm)      |              |                              |
| Root Distributions                                 | Χ                                    | Х            |                              |
| %C and %N                                          | X                                    | Χ            | Χ                            |
| Extractable nitrate concentrations                 |                                      | Х            |                              |
| $\delta^{15}N$                                     |                                      | Χ            |                              |
| рН                                                 | X                                    | Χ            | X                            |
| Effective cation exchange capacity (ECEC)          | Χ                                    |              |                              |
| Texture                                            | X                                    |              |                              |
| Wet aggregate size distribution (ASD)              |                                      | Х            |                              |
| Dissolved organic carbon (DOC)                     |                                      | Х            | X                            |
| Microbial biomass carbon                           |                                      | Х            |                              |
| β-glucosidase and N-acetyl-β-D-<br>glucosaminidase |                                      | Х            |                              |

240 241 3.1 Measuring Soil Organic C and Nitrogen Dynamics 242 We assessed SOC and SON concentrations and stocks and the likelihood of SOC and SON 243 degradation by microbes by analyzing bulk soil samples at 10-cm intervals. We determined SOC 244 and SON on subsamples (~75 mg) via an elemental analyzer (Vario Macro Cube, Elementar, 245 Ronkonkoma, NY). We used SOC and SON concentration measurements to calculate each 246 subsample's C:N ratio. To determine stocks of SOC in each horizon, we multiplied SOC 247 concentrations by soil bulk density obtained in each horizon. Bulk density was measured using a 248 three-dimensional laser scanner (3D Scanner Ultra HD, NextEngine, Inc., Santa Monica, CA) 249 following Rossi et al. (2008). We measured extractable, dissolved organic C (DOC) to estimate organic C that can be 250 251 relatively easily mobilized and transported out of the soil profiles; note that this differs from 252 DOC measured in soil porewater using lysimeters, and instead represents a salt-extractable 253 pool. We analyzed soil samples at 10-cm intervals to auger refusal collected at each site during 254 the growing season. Soil samples were extracted within three months of collection date. A total 255 of 7.5 g of soil at field moisture was extracted with 30 ml of simulated rainwater (Laegdsmand et al., 1999). The extracted soil solutions were comprised of 47.9  $\mu$ M NaNO<sub>3</sub>, 4.69  $\mu$ M KCl, 256 257 23.81 μM CaCl<sub>2</sub> x  $2H_2O$ , 12.09 μM MgSO<sub>4</sub> x  $7H_2O$ , and 18.24 μM (NH<sub>4</sub>)<sub>2</sub>SO<sub>4</sub> and adjusted to a pH 258 of 4.2 ± 0.5 using HCl. Samples were placed on a shaker table for 30 minutes and centrifuged at 259 80 Hz (s<sup>-1</sup>) for 15 minutes. Samples were filtered through 0.45 μm nylon syringe filters and 50 260 ml acid washed syringes. Filtered samples were stored in 10 ml centrifuge tubes, frozen and 261 shipped overnight in a cooler with dry ice to the University of Kansas. DOC was analyzed from 262 the thawed samples using a Violet-pink Mn (III)-pyrophosphate solution and a microplate 263 reader (Biotek, UT). 264 To better understand the potential for microbial activity in these soils, we quantified microbial biomass C by horizon from pits dug in the summer of 2022 (Brooks et al. 1985). We exposed 5 g 265 266 of each soil sample to chloroform for 24 h. To these fumigated sub-samples and to 5 g of 267 unfumigated sub-samples, we added 20 ml of 0.5 M K<sub>2</sub>SO<sub>4</sub> and shook for 30-40 minutes at 220 268 rpm. These samples were filtered through a 0.45 μm polyethersulfone (PES) filter and their DOC 269 concentration was determined via colorimetry (Bartlett and Ross 1988) on a Synergy HT 270 microplate reader (Agilent, USA). To assess the degree to which soil microbial communities 271 were generating exo-enzymes that catalyze soil organic matter decay and thus can provide 272 assimilable C- and N-rich compounds, we quantified potential activity rates of two such 273 enzymes. We measured activity of β-glucosidase and N-acetyl-β-D-glucosaminidase, herein 274 referred to as BGase and NAGase, which are linked to microbial C (BGase) and N and C 275 (NAGase) acquisition (Sinsabaugh and Moorhead, 1994; Allison et al., 2011, Stone et al., 2014), 276 using 4-methylumbelliferyl β-D-glucopyranoside (for BGase) and 4-methylumbelliferyl N-acetyl-

- β-D-glucosaminide (for NAGase) fluorescent tags. These tags were added to slurries made from
- approximately 1 gram of soil and pH-adjusted 50 mM sodium acetate. We pipetted the blended
- sample into the desired substrate and incubated all plates at 25 °C for 18 hours. Fluorescence
- from a Synergy HT plate reader (Agilent, USA) was used as a proxy for each enzyme's capacity
- to cleave monomers from the respective molecules undergoing decay (DeForest, 2009; German
- et al., 2011).
- We quantified salt-extractable NO<sub>3</sub><sup>-</sup> because of its importance as a biotically-available form of
- N, and also because of its status as a readily leachable ion. As such, it can serve as an indicator
- of each soil's capacity to undergo elemental loss in surface soil with hydrologic fluxes, and
- provides a valuable point of comparison to DOC values. We extracted ~10 g (fresh weight) of
- each soil sample with 0.5M K<sub>2</sub>SO<sub>4</sub> and repeated the shaking and filtering steps described above
- for MBC. Extracts were analyzed for NO<sub>3</sub><sup>-</sup> (Synergy HT, Agilent, USA) using Shand et al. (2008), a
- microplate-based approach that relies on hydrazine sulphate and sulphanilamide to generate a
- color intensity directly related to NO<sub>3</sub>- concentration.
- We also quantified soil organic matter  $\delta^{15}N$ , given these signatures' value as an indicator of the
- degree to which soil microbes have processed soil organic matter (Nadelhoffer and Fry 1988;
- Billings and Richter 2006). Sub-samples of each soil were dried, ground to fine powder, and
- weighed into a tin capsule for analysis. Values of  $\delta^{15}N$  were obtained at the Kansas State
- University Stable Isotope Lab, where an Elementar EA Vario Pyrocube linked to an Elementar
- GeovisiON isotope Ratio Mass Spectrometer determine N concentration and  $\delta^{15}$ N, respectively.
- 3.2 Measuring Soil Chemical and Physical Properties
- To better assess possible differences in the chemical and physical controls on SOC stability we
- also measured pH, effective cation exchange capacity (ECEC), soil texture, and wet aggregate
- size distribution (ASD). We focused on pH as it is known to strongly control microbial
- communities and mineral associated organic C (MAOC) (Kleber et al., 2015). The soil pH was
- determined in a 1:1 H<sub>2</sub>O soil slurry (Soil Survey Staff, 2022). We focused on ECEC because ECEC
- has a high positive correlation with SOC, clay content, and aluminum and iron oxides (Solly et
- al., 2020), which are highly correlated with the formation of MAOC (Kleber et al., 2015). ECEC
- was determined by summing Ca, Mg, and K extracted using a Mehlich-3 solution (Culman et al.,
- 2019). Mehlich-3 extraction was used instead of an ammonium acetate extraction, because the
- soils samples had a pH of <7.5 and there is very little to no calcium carbonate. In these
- conditions Mehlich-3 and ammonium acetate extractions yield similar ECEC values (Rutter et
- al., 2021).
- We examined soil texture at each pit for several reasons. First, the total amount of clay is
- important to MAOC, and second, texture is known to impact the distribution and connectivity
- of pores. This connectivity influences how easily oxygen can diffuse into a soil profile and thus

processes such as microbial respiration (Schjønning et al., 1999; Moldrup et al., 2001), and 314 further regulates water and solute transport down-profile. Soil texture was analyzed on pit 315 samples collected from 2020-2021 using a laser diffraction (LD) unit (Bettersizer S3, Bettersize 316 Instruments, Dandong, Liaoning, China). Five grams of soil was sieved to 2 mm, and organic 317 matter was removed by treating samples with 30% hydrogen peroxide. Ten ml of 10% sodium 318 hexametaphosphate (HMP) was added to the solution to prevent flocculation. The soil solution 319 was pipetted into the Bettersizer until obscuration levels were between 14-20. We set clay-silt 320 and silt-sand boundaries to be 6.6 and 60.33 μm, respectively (Makó et al., 2017).

We quantified aggregate size distributions as one key metric of soil structure. Aggregate-size distributions were measured on each soil horizon following Nimmo and Perkins (2002). Briefly, around 25 g of the largest air-dried aggregates were fully saturated with a Dickson apparatus (Dickson et al., 1991), and placed on a Yoder device where sieves (#4, 10, 17, 70) and soil samples were raised and lowered in the water 2.8 cm per stroke at a rate of 36 strokes per a minute for 10 minutes. Following this agitation in water, the sieves with their respective aggregates were placed in a drying oven at 105 °C for 12 hours. The soil material remaining on each sieve was dispersed with 200 ml of 2 g L<sup>-1</sup> HMP, mixed for 10 minutes, passed through the sieve again, and oven-dried at 105 °C for 2 hours. Weights were recorded and mass fractions of water-stable aggregates were then calculated. Sieves divided aggregates into 5 classes: aggregates > 4.76 mm, aggregates between 2-4.76 mm, aggregates between 0.21-1 mm, and aggregates less than 0.21 mm. To simplify our analysis, we agglomerated these into 3 classes following Souza et al. (2023): fine aggregates (< 0.21 mm), intermediate aggregates (0.21–4.76 mm), and coarse aggregates (> 4.76 mm). A weighted geometric mean aggregate diameter (GMD) was calculated for each triplicate using the mass fractions of each aggregate-size class; the mean and standard deviation were calculated from these triplicate values to represent the aggregate diameter of each sample. The GMD values were divided by SOC content and the resulting values were used to characterize the propensity of C to form aggregates.

## 3.3 Measuring Rooting Distributions

327328

To determine the relationship between roots, and C stability and transport, we measured the fraction of soil volume containing fine and coarse roots throughout the soil profiles using images collected from all 10 pits (e.g., 2020/2021 and 2022). We used ImageJ (Schneider et al., 2012) to overlay each image with a 1 cm x 1 cm grid. We then manually checked each 1 cm x 1 cm grid cell for the presence of a fine root (diameter < 1 mm) or coarse root (diameter > 1 mm) and noted these presence/absence scores for each grid cell. Our focus is the soil volume containing roots and thus directly influenced by roots. As such, only presence/absence and not count data were recorded, and in any cell containing both fine and coarse roots the presence of only the coarse root(s) was recorded given their greater volume (Billings et al., 2018). These measures are thus a conservative measure of direct root influence on soil volumes, derived at

the cm scale for soil pedons. Centimeter-scale cell presence/absence data were transformed into the fraction of each 1-cm thick layer containing roots.

### 3.4 Sensor Data

363364

368369

Soil sensor arrays were installed in the first set of pits (2020/2021) at the completion of sampling. Sensors were installed at depths of 15 cm, 45 cm, and 110 cm (or deepest depth; Table 1) to monitor soil temperature (°C) and volumetric water content (VWC; EC-5, Meter Group, Pullman, WA), matric potential (kPa) (Teros 21, Meter Group, Pullman, WA), O<sub>2</sub> concentration (%) (IB201806, Apogee Instruments, Logan, UT) and CO<sub>2</sub> concentration (ppm) (F0275476, Eosense, Dartmouth, Canada). Data were collected every 30 minutes for moisture, matric potential, and temperature and hourly for  $O_2$  and  $CO_2$  given the power requirements. We focus on CO<sub>2</sub> and O<sub>2</sub> as they are indicators of soil microbial and root biotic activities including heterotrophic respiration. Microbial activity directly and indirectly affects the formation of MAOC, SOC stabilization, and microaggregation (Dohnalkova et al., 2022). We used sensor data to investigate additional environmental controls on C dynamics. We converted O<sub>2</sub> from millivolt readings to % by adding calibrated values to the millivolt value of O<sub>2</sub>. Each calibrated value was specific to the sensor installed and determined using atmospheric concentrations prior to installation. To focus on the growing season, we selected data from June 15-August 29, which was 14 days before the first sample was collected (June 29th) and ending 14 days after the last sample was collected (August 15th). AS reflects 2021 data, SS reflects average daily 2021 and 2022, and ESG, EAG, and SG reflects 2022 data. These differences were because pits were installed with sensors in different years and some of the instrumentation had power outages and other unforeseen issues. We averaged daily temperature and VWC by week and examined average and standard deviation of the O2 and CO<sub>2</sub> over the growing season.

375

376377

### 3.5 Data Analysis

Spatial replicates controlling for all ecosystem-scale factors were not feasible in this study. Instead, we advance our understanding of SOC stability by examining a more diverse suite of biotic and abiotic ecosystem characteristics than is often the case in SOC-focused work. Our work begins to unravel the complex interactions among cover type characteristics, soil properties, and hydrologic settings in SOC dynamics. We used Wilcoxon Rank sum tests to determine if differences between aspen and spruce concentrations of SOC, DOC, total soil nitrogen and nitrate, and of ratios of DOC to SOC were significant. We used linear mixed effects (LME) methods via the R package lme4 (Bates et al., 2014) to assess the influence of vegetation type, depth, and their interaction (N=5) on soil abiotic conditions, various forms of soil nitrogen and C and  $\delta^{15}$ N, ASD, and root abundances. We tested if variables were normally distributed using the Shapiro-Wilks test and transformed the data to achieve a normal distribution if they were non-normal. The soil chemical properties of SOC, EOC, EOC:SOC, ECEC were log

transformed, while C:N data were transformed with the function  $x^{1/3}$ . Root fractions and soil solution pH did not require transformation to meet model assumptions. We assessed if vegetation type exerted a meaningful influence on the previously mentioned variables by constructing four models. The two simplest models included only vegetation type or depth, both as fixed effects. A third model included those fixed effects additively (e.g., Vegetation + Depth), and a fourth model included their interaction. We resolved the lack of independence of soil depth within each pedon by incorporating site identifiers as a random effect term in the model. We then tested the normality of the model residuals using the Shapiro-Wilk test. For all models that passed this test, we compared the model fits using analysis of variance (ANOVA) and visually examined model residual errors for homogeneity of variance; the best model fit was selected based on the lowest Akaike information criterion (AIC) following Hauser et al. (2020). We interpret the results of these LME models conservatively, given the low number of replicate sites for each land cover type. We could not perform a LME model on microbial biomass and enzyme data due to the relatively limited number of samples. This limited our ability to include vegetation\*depth interactions in those models.

### 4 Results

### 4.1 Soil Properties and Development

- Clay, silt, and sand content at the aspen sites (AS and EAG; Fig. S1) and one of the conifer sites (ESG) exhibited little variation with depth (average 33.1% clay and 18.8% sand), while the two other conifer sites had a greater sand and lower silt and clay content, particularly at depths greater than 25 cm (SS and SG; Fig. S1). Cation exchange capacity was similar among the aspen and conifer sites with averages of  $7.2 \pm 4.9$  and  $8.2 \pm 6.8$  (meq/100 g soil), respectively, with elevated values at the surface that declined with depth (Fig S2).
- We were able to access and describe soil profiles to approximately 100 cm (Fig. 2, Table. S1). All sites had weak to moderately strong subangular blocky structure throughout the soil profile, and most sites had weak to moderately strong granular structure in A and upper B horizons. Dendritic tubular pores, interpreted to be abandoned root channels, were present throughout the soil profile of aspen sites, while they were less common in the conifer soil profiles. Aspen sites exhibited faint organic stains and organoargillans (i.e., dark, organic stained clay films) throughout the soil profile, while conifer sites had clay bridges and krotovina throughout the soil profile. The krotovina suggest greater bioturbation under conifer than aspen. Both vegetation types exhibited ferriargillans (i.e., clay coats that include Fe oxides), clay films, and charcoal, although ferriargillans and clay films were more prominent under conifer. Clay bridges, organoargillans, and ferriagllians indicate illuviation. Lithologic discontinuities were identified in SS, ESG, and EAG indicating colluvial inputs into these footslope pedons.

Figure 2: Soil profiles described at each site. Horizon colors represent the moist color of the soils as matched to the soil-color or Munsell chart.

Soils at both aspen sites (AS and EAG) are Ustic Haplocryolls with thick, SOC-enriched surface horizons (mollic epipedons) and showing evidence of incipient subsoil development in the form of moderately thick cambic horizons. Soils under conifer sites are Typic Haplocryepts (SS and SG) and Eutric Haplocryalfs (ESG). Although surface horizons under conifer were not as well-developed (ochric epipedons), the subsurface showed similar incipient pedogenesis in the form of cambic horizons for SS and SG and greater development in the case of ESG where an argillic horizon was identified between 19-90 cm below the mineral surface.

### 4.2 Soil abiotic conditions

To understand how soil abiotic conditions are linked to SOC forms and processing pathways, we focused our analysis of soil temperature and moisture during the growing season (June – August; Fig. 3). As expected, soil temperature increased at all sites as the growing season progressed peaking in mid to late July, with the warmest temperatures observed near the surface and lower variability observed at depth. We also observed that the aspen sites (AS & EAG), which are on south-facing slopes, are warmer than conifer sites with an average surface soil (15 cm deep) temperature of  $14.3 \pm 1.2$  and  $10.4 \pm 1.0$  °C, respectively, during the growing season. Aspen sites were generally drier than conifer. The average volumetric water content in the surface soils (15 cm deep) at aspen sites was  $0.15 \pm 0.05$  and the average volumetric water content at spruce sites was  $0.24 \pm 0.05$  cm<sup>3</sup> cm<sup>-3</sup>.

We also examined soil pH. Across the entire soil profile, pH was similar at the aspen and spruce sites  $(5.6 \pm 0.3 \text{ and } 5.3 \pm 0.4, \text{ respectively})$  but their depth trends differed with spruce soils having slightly more acidic pH near the surface compared to the aspen (Fig. S2). This trend reversed at approximately 60 cm, where the aspen soils became slightly more acidic compared to the conifer soils.

Figure 3: Temperature (a-e; °C) and volumetric water content (VWC (cm³ cm⁻³); f-j) data for aspen sandstone (AS; a, f), spruce sandstone (SS; b, g), spruce granite (ESG; c, h), aspen granite (EAG, d, i), and spruce granite (SG; e, j). AS reflects 2021 data, SS reflects averaged 2021 and 2022, ESG, EAG, and SG reflects 2022 data.

### 4.3 Soil Organic C and Nitrogen

Across all sites, SOC concentrations ranged from 46.0-62.6 mg g $^{-1}$  near the surface (5 cm deep) to 4.8 to 29.0 mg g $^{-1}$  at depth (Fig. 4a). SOC concentrations were generally higher under aspen compared to spruce sites (p 

Figure 4: a) Soil organic carbon (SOC) concentrations, (b) SOC stock [by horizon per pit; solid lines indicate sites underlain by granite and dotted lines aspen], (c) dissolved organic carbon (DOC), and (d) the ratio of dissolved organic carbon to soil organic carbon (DOC:SOC) with depth under two different vegetation types, aspen (orange) and spruce (green) at the Coal Creek catchment, Colorado, USA. Values represent mean +/- standard deviation.

Total soil nitrogen ranged from 0.2 mg g<sup>-1</sup> at depth to 4.63 mg g<sup>-1</sup> near the surface. A model including an interaction between vegetation type and depth was the best fit (p = 0.003; Fig. 5a). Aspen values were greater than those in spruce-dominated soils at all depths; the significant interaction implies that the decline with depth was greater in spruce soils. Nitrate concentrations averaged 214  $\pm$  323 ng g<sup>-1</sup> near the surface and 2.3  $\pm$  3.8 ng g<sup>-1</sup> at depth. Nitrate was elevated under the aspen compared to the spruce sites (Fig. 5b), and values under both vegetation types varied with depth. A model that included both depth and vegetation type with no interaction was a meaningfully better predictor of nitrate concentrations than either depth or vegetation alone (p = 0.035), and including a depth-vegetation interaction did not improve model fit. Aspen soil C:N averaged 10.9 ± 1.1 and remained fairly constant with depth (Fig. 5c). The spruce sites showed greater variation with depth with a similar mean value of 19.3 in the top 20 cm but widely variable values at the deepest points, ranging from 4.6 to 28.7 (Fig. 5c). Including the interaction between vegetation type and soil depth improved model fit (p = 0.0008), indicating that C:N varied more with depth in spruce soils than in aspen soils, where values stayed fairly constant. The lowest C:N value, found at depth in one of the spruce forest, suggests that the soil organic matter there has been heavily processed by microbes (Ziegler et al. 2017).

 $\delta^{15}$ N signatures showed less distinct depth trends compared to the total nitrogen and nitrate, mirroring the relative lack of clear depth trends in C:N. Though variation across sites limited our ability to find statistically-significant differences across vegetation types or a significant influence of depth,  $\delta^{15}$ N of soil organic matter in spruce plots tended to be lower than that of aspen (Fig. 5d), hinting that soil N has undergone more microbial processing (Nadelhoffer and Fry 1988; Billings and Richter 2006) under aspens compared to under conifers. This

interpretation is consistent with the mean spruce C:N values being greater than those in aspen forests (Fig. 5c).

Figure 5: (a) Soil nitrogen, (b) soil nitrate, (c) carbon to nitrogen ratio (C:N), and (d)  $\delta^{15}$ N with depth under two different vegetation types, aspen (orange) and spruce (green) at the Coal Creek catchment, Colorado, USA. Values represent mean +/- standard deviation. For each mean and standard deviation, where error bars are not visible the deviation is smaller than the point.

## 4.4 Biotic activity

### 4.4.1 Roots

The LME models indicate that models including vegetation-depth interactions were the most effective at describing total and coarse root fractions (p <0.001), with generally greater root abundances in the aspen compared to the spruce (Fig. 6a & c). In contrast, vegetation type offered no additional explanatory power to the depth-dependent fine root abundance (p > 0.05; Fig. 6b), suggesting that the greater total root abundance under aspen was driven more by the coarse root fraction. The difference between aspen and spruce root abundances were continuous with depth for the total root fraction but more punctuated with coarse root fraction. For example, higher coarse root fractions were observed from 30-60 cm and greater than 90 cm for the aspen as compared to the spruce. Interestingly, overall spruce root fractions decreased faster with depth than aspen root fractions. When we standardized DOC with rooting abundance, we found generally greater concentrations of DOC per unit root abundance under spruce soils, particularly with respect to total and fine roots (Fig. 7).

Figure 6: (a) Total, (b) fine, and (c) coarse root fractions quantified at 1-cm depth interval under two different vegetation types, aspen (orange) and spruce (green) at the Coal Creek catchment, Colorado, USA. Values represent mean (points) +/- standard deviation (shading).

Figure 7: DOC divided by mean (a) total, (b) fine, and (c) coarse root fractions every 10 cm under two different vegetation types, aspen (orange) and spruce (green) at the Coal Creek catchment, Colorado, USA. Values represent mean (points) +/- standard deviation (bars). Root fractions represent the count of fine (

Figure 8: Soil (a)  $\beta$ -glucosidase (BGase) (b)  $\beta$ -N-acetyl glucosaminidase (NAGase) (c) microbial biomass and (d) the ratio of BGase to NAGase at the Coal Creek catchment, Colorado, USA. Each point represents one site and one depth, and curves represent exponential fit of the data, with each curve defined by multiple spruce and aspen sites.

## 4.4.3 Soil $O_2$ and $CO_2$

We examined soil  $O_2$  and  $CO_2$  concentrations during the growing season to better understand patterns of respiration (Fig. 9). Soil  $CO_2$  concentrations increased with depth across all sites, while  $O_2$  concentrations were more variable. Soil  $O_2$  concentrations remained relatively stable at aspen sites and at spruce granite sites (AS, EAG, and SG). However,  $O_2$  concentrations decreased with depth at the remaining two spruce sites—one sandstone and one granite (SS and ESG).

Figure 9: Soil gas concentrations of  $O_2$  (%) and  $CO_2$  (%) at aspen (orange) and spruce (green) sites during the growing season at depths 15 cm (light), 45 cm (medium), and 110 cm (dark) at the Coal Creek catchment, Colorado. Values represent mean (points) +/- standard deviation (bars) with lines connecting depths within each profile. The shallowest depth of each site is labeled: AS, Aspen Sandstone; EAG, Elk Aspen Granite; ESG, Elk Spruce Granite; SG, Spruce Granit; SS, Spruce Sandstone.

## 4.4 Soil Aggregates

The mean geometric diameter of soil aggregates was generally smaller under aspen compared to spruce (Fig. 10a). Aspen aggregates tended to be finer, with fewer intermediate and coarse aggregates compared to the spruce soil (Fig. 10b-d). LME models indicated that vegetation-depth interactions were the most meaningful in driving all three aggregate size classes (p 

Figure 10: (a) Diameter of soil aggregates, and fraction of (b) fine aggregates (< 0.21 mm), (c) intermediate aggregates (0.21-4.76 mm), and (d) coarse aggregates (> 4.76 mm) that contribute to the overall soil aggregate diameter at the Coal Creek catchment, Colorado, USA. Values represent (a) geometric mean or (b-d) mean (points) +/- standard deviation (bars). Colors are associated with vegetation, aspen (orange) and spruce (green).

Figure 11: The fraction of (a) fine aggregates (< 0.21 mm) and (b) coarse aggregates (> 4.76 mm) divided by the fraction of SOC at the Coal Creek catchment, Colorado, USA. Values represent mean (points) +/-standard deviation (bars). Colors are associated with vegetation, aspen (orange) and spruce (green). Please note each aggregate size class is divided by the total SOC, not the C associated with each size class.

### 5 Discussion

589

599

By integrating knowledge from biology, pedology, hydrology, and soil chemistry we were better able to understand how multiple factors interact to drive observed SOC patterns in aspen and conifer montane forests. Our data indicate that differences in SOC protection contribute to the commonly observed patterns of elevated SOC storage in soils beneath aspen compared to conifer stands (Woldeselassie et al., 2012, Laganiere et al., 2013, Boča et al., 2020, Román Dobarco et al., 2021). Furthermore, our study suggests that aspen-dominated soils may experience enhanced degrees of microbial transformation of SOC, with the products of those transformations exhibiting a greater tendency to reside in relatively small aggregates and thus protect C to a greater degree (Fig. 12). Consistent with this idea, we also observed less DOC loss in aspen soils compared to soil under spruce stands and slightly higher concentrations of DOC per unit root abundance under the spruce stands. These differences suggest greater infiltration of DOC to deeper horizons in spruce soils compared to those in aspen stands. It is important to highlight that spatial replicates controlling for all factors of interest at an ecosystem scale were not feasible, but that our work moves beyond considerations of vegetation biomass characteristics that often dominate investigations of contrasting SOC dynamics. Instead we begin to unravel the complex interactions among cover type characteristics, soil properties, and hydrologic settings (e.g., hillslope position). Below, we discuss the drivers of SOC form and fate in greater detail and interpret these findings in light of recent increases in stream water DOC concentration in this spruce-dominated watershed.

Figure 12. Summary of observations across aspen and spruce sites at Coal Creek, CO (USA) that are interpreted to indicate a greater amount of chemical and physical protection of SOC under aspen sites.

5.1 Microbial data are consistent with the Microbial Efficiency - Matrix Stabilization framework

Our data provide multiple lines of evidence that SOC protection, and thus C fate, in these

montane forests is largely controlled by biotic processes linked to soil mineral material. Here

greater values of total N, nitrate, BGase and  $\delta^{15}$ N and lower C:N under aspen compared to

spruce (Fig. 5, 8), suggest a greater degree of microbially processed organic matter under the

aspen stands where greater SOC contents were measured (Fig. 4). These data hint that the

618 microbial community under aspen stands functions in a manner consistent with the Microbial

Efficiency - Matrix Stabilization (MEMS) framework (Cotrufo et al., 2012), transforming

relatively labile leaf litter (e.g., under aspen) into byproducts more readily stabilized within soil

profiles to a greater extent than appears to occur with slower-turnover litterfall (e.g., spruce).

Differences in litterfall composition and thus decay rates across aspen and conifer species have

been widely reported, with generally lower lignin and higher nitrogen content in aspen litter

(Moore et al., 2006). Our inference about litterfall differences promoting microbial byproduct

stabilization is consistent with findings from across western Canada, where investigators

observe relatively more active microbial communities under aspen compared to paired spruce

stands throughout a growing season (Norris et al., 2016). Specifically, one interpretation of

these C:N,  $\delta^{15}$ N, exo-enzyme, SOC, and DOC data at our sites is that tree species-specific

composition of litterfall appears to have prompted greater microbial activities (Fig. 8), likely

promoting greater contributions of microbial necromass to the SOC pool. This, in turn, may

promote greater SOC retention in aspen-dominated soils; though investigation of specific

necromass-derived compounds in these soils (e.g., Liang et al., 2019) is beyond the scope of this

work, it represents a valuable way forward to testing this inference.

5.2 SOC transformations likely influence aggregate sizes and the probability of destabilization

The smaller aggregate sizes in aspen-dominated soils further support the notion that SOC

stability is enhanced by higher microbial activity and increased necromass production rates.

SOC is better protected and has generally longer mean residence times in smaller aggregates

than larger aggregates (Six and Jastrow, 2002; Six et al., 2004). Literature hints that the larger

size aggregates (Fig. 10c) and greater propensity for C to form large aggregates (Fig. 11b)

observed in the spruce-dominated soils at our sites may be due to a greater abundance of

particulate organic matter (POM) in spruce compared to aspen forest soils (Cotrufo et al., 2015;

Cotrufo et al., 2019); this may be the case since spruce litterfall is more difficult to decompose.

Taken together, these lines of evidence are consistent with aspen-dominated forests harboring

SOC pools that tend to promote relatively small aggregate formation that can preserve SOC to a

greater extent — especially at depth, where MAOC tends to dominate SOC pools (Jackson et al.

2017).

614

Our soil data also indicate that SOC pools in aspen soils are more strongly dominated by MAOC

compared to those in spruce soils. The greater abundances of smaller aggregates and total soil

of nitrogen and nitrate concentrations, and lower C:N ratios (Fig. 5a-c), in aspen compared to

conifer soils are consistent with relatively greater MAOC than POC concentrations (Kögel-

Knaper et al., 2008; Ye et al., 2018; Sokal et al., 2022). Combined with the lower DOC:SOC ratio

in aspen-dominated soils, these data suggest that a greater fraction of SOC in aspen-dominated

soils is mineral-bound and relatively difficult to transform into microbially-available pools of

654 DOC. We interpret these data to suggest that microbially-mediated transformations of SOC

promote differences in the abundance of MAOC and the physical structure of soil aggregates 656 that leads to differences in the SOC protection.

### 5.3 Roots may indirectly regulate depth profiles of EOC losses

664

676

685

692

Roots can influence SOC stability through their promotion of both physical and chemical protection. Specifically, roots can play an important role in the formation and breakdown of soil aggregates (Oades, 1984; Singer et al., 1992; Le Bissonnais 1996; Attou et al., 1998), they can create biopores that can support the transport of DOC in deeper soil layers (Sigen et al., 1997; Angers and Caron, 1998; Boger et al., 2010; Zhang et al., 2015; Lucas et al., 2019), and root exudates can prime microbial activity, enhance decomposition, and support the formation of MAOC (Jilling et al., 2021; Fossum et al., 2022). Our data revealed little direct correspondence of root abundance with SOC. However, per unit root abundance, spruce soils appear to harbor more DOC compared to aspen (Figure 7). This pattern—especially evident in total and fine root abundance—suggests that DOC moves more readily through spruce soil profiles, potentially leading to greater DOC losses to stream water compared to aspen-dominated soils. A complementary explanation would be that there are differences in the amount of DOC exudation by roots between the two species, and indeed such difference in exudation rates have been hypothesized in the literature (Buck and St. Clair, 2012; Boča et al., 2020). We might expect that greater exudation would lead to a greater increase in the MAOC pool and enhanced C stability (Even and Cotrufo, 2024), which could explain the lower values of DOC relative to SOC observed under aspen.

### 5.4 Aspect exerts some control on Coal Creek SOC dynamics.

South-facing slopes tend to be warmer and drier than north-facing slopes in the northern hemisphere (Burnett et al., 2008), and thus they can prompt more microbial decomposition of SOC. In our study aspen cover occurs where soil temperatures are warmer (Fig. 2). As such, it is possible that the exo-enzymatic signals of generally greater microbial activity in aspendominated soils compared to spruce-dominated soils (Fig. 8) is prompted more so by enhanced soil temperatures than by differences in aspen and spruce organic matter characteristics, and that enhanced soil temperatures also contribute to smaller soil aggregates, perhaps also due to greater microbial activities. We note that the volumetric fraction of soil moisture was also lower in the aspen, particularly at the shallowest soils, but that aspen soils appear to stay sufficiently moist (0.10-0.20 under aspen vs. 0.20-0.30 under the spruce) to support microbial activity responses to the higher temperatures. Consistent with this idea, soil CO<sub>2</sub> and O<sub>2</sub> concentrations generally suggest that microbial activities in the warmer, aspen-dominated soils are greater than in the cooler, spruce-dominated soils. Cooler, wetter conditions of the sprucedominated soils, particularly following snow melt may prompt a deeper infiltration of moisture and DOC down the soil profile, leading to the elevated DOC/root biomass observed under the spruce stands. While disentangling the impact of elevated soil temperatures from that of the chemical composition of organic inputs from aspen trees within the soil profile is difficult, soil nitrogen and  $\delta^{15}N$  data are consistent with the idea that litterfall chemistry, and not just temperatures, promoted greater microbial activities in the aspen-dominated soils. We suggest that investigating the comprehensive, integrated effects of warmer, aspen-dominated sites on SOC dynamics compared to cooler, spruce-dominated sites offer a straightforward approach to assessing landscape-scale transitions in watershed C dynamics.

698 6 Changes in SOC destabilization and release have implications for stream water quality 699 Widespread increases in stream water DOC concentrations have been reported around the 700 world in recent decades (Evans et al., 2005; Alvarez-Cobelas, 2012; Stanley et al., 2012; Pagano 701 et al., 2014). Increases in stream water DOC concentration can harm global water quality by 702 altering light and thermal regimes, nutrient cycling (e.g., Morris et al., 1995; Cory et al., 2015), 703 the transport and bioavailability of heavy metals (e.g., Dupré et al., 1999; Trostle et al., 2016), 704 and creating harmful disinfection byproducts (Leonard et al., 2022). Consistent with these 705 global trends, recent findings at Coal Creek also report increasing DOC concentrations (Leonard 706 et al., 2022; Kerins et al., 2024). As such, our research may help to shed light on drivers of 707 stream water DOC, and thus has implications for changing drinking water quality in the region. 708 Specifically, our work hints that differences in aggregate-size distributions may play an 709 underappreciated role in influencing stream water C chemistry. Aggregate size can be 710 modulated by vegetation type (i.e., smaller aggregates associated with Aspen) (Fig. 11; Jiménez 711 et al., 2012; Zhao et al., 2017), and aggregation and disaggregation both represent mechanisms 712 that can influence the transport of DOC to streams (Fan et al., 2022). Larger aggregates appear 713 more prone to induce DOC transport into streams due to their relatively greater propensity to 714 undergo fragmentation and associated loss of DOC (Cincotta et al., 2019; Fan et al., 2022). 715 Understanding how these different types of vegetation affect the chemical and physical 716 properties of soil, and how this influences C release, is further complicated by climate change. 717 Increasing temperature, a phenomenon evident in many Rocky Mountain environments 718 including Coal Creek (Zhi et al., 2020), can cause aggregates to become less stable (Lavee et al., 719 1996, Wang et al., 2016), soil microbes to increase their C demand (Belay-Tedla et al., 2009, Hu 720 et al., 2017), and recalcitrant C to undergo decay more rapidly (Luo et al., 2009). Dry soil 721 conditions, which are often prompted by warming (Lakshmi et al., 2003), can induce a decrease 722 in microbial biomass, which is often incorporated into stable aggregates (Gillballi et al., 723 2007). In addition to warming induced changes to subsurface properties and function, changing 724 stand composition prompted by warming and drying can alter C dynamics. Some research 725 indicates a high mortality rate among aspen stands and the expansion of conifer stands 726 associated with increases in drought (Anderegg et al., 2013, Brewen et al., 2021), while others 727 indicate the expansion of bark beetles and wildfires may promote the encroachment of aspen 728 into conifer stands (Andrus et al., 2021). Our work suggests that the distribution of spruce and 729 aspen in a watershed may influence soil release of DOC and its subsequent transport into 730 streams, given that spruce vegetation appears to be associated with larger aggregates (Fig. 10), 731 a potential for greater DOC loss per unit SOC (Fig. 4c), greater sand content (depth >25 cm; Fig. 732 S1) and thus likely greater values of hydraulic conductivity, and generally higher soil moisture content (Fig. 3) compared to aspen-dominated soils. Thus, shifts in stand composition 733 734 associated with perturbations linked to large-scale global changes have the potential to 735 influence DOC transport from the hillslope to the stream.

### 7 Conclusions

Our work explores the interplay of different forest cover types and abiotic conditions in 739 governing soil microbial activities, which then influence the propensity of SOC pools to form 740 and stabilize soil aggregates of different sizes. In turn, these processes appear to promote 741 varying capacities of a soil to protect SOC from destabilization. Our work contributes to the 742 ongoing efforts to investigate suites of biotic and abiotic features across whole ecosystems — 743 the critical zone (Richter and Billings, 2015)—to better understand SOC dynamics (e.g., Keller, 744 2019; Mainka et al., 2022; Wasner et al., 2024). It also provides a foundation for future studies 745 that could incorporate more spatially replicated sites across key environmental gradients. 746 Specifically, our data suggest that organic matter from aspen supports higher microbial 747 transformation rates and greater stabilization of SOC, reducing the likelihood that labile SOC is 748 transported down the soil profile. Consequently, aspen-dominated stands may be less likely to 749 promote the movement of DOC across the landscape and into streams. This phenomenon may 750 be driven by greater rates of microbial necromass formation and generation of relatively 751 smaller aggregates, and highlights how models like MEMS (Cotrufo et al., 2013) can be 752 important for projecting not just CO<sub>2</sub> release to the atmosphere and SOC stabilization, but 753 down-profile and downstream C transport as well. Though soil temperature differences likely 754 played a role in the greater soil microbial activities in aspen, the generally higher nitrogen in 755 aspen soils lends credence to the idea that litterfall chemistry itself played a key role in the 756 higher rates of soil microbial activities. As such, the patterns that emerge in our data suggest 757 that processes that control landcover ultimately also control SOC dynamics and soil structure in 758 ways that may directly impact the delivery of organic C pools deep within soil profiles and 759 stream water quality, and be sensitive to changing climatic conditions. Here, we demonstrate 760 how the critical zone paradigm offers a valuable approach for examining, interrogating, and 761 understanding watersheds, linking vegetation dynamics to subsurface processes and ultimately 762 to the flux of water and C from hillslopes to streams.

### 763 Data statement

- Soil sensor and soil properties data can be obtained at HydroShare,
- http://www.hydroshare.org/resource/9948ad04a9a74246ad9bd5f8decb40b9

| 766        |                                                                                                                                                              |
|------------|--------------------------------------------------------------------------------------------------------------------------------------------------------------|
| 767        | Author Contributions (CRediT):                                                                                                                               |
| 768<br>769 | Wang, L: Conceptualization, Data curation, Formal analysis, Investigation, Methodology, Visualization Writing – original draft, Writing – review and editing |
| 770<br>771 | Billings, S.: Conceptualization, Funding acquisition, Investigation, Methodology, Writing – original draft, Writing – review and editing                     |
| 772        | Li, L.: Conceptualization, Investigation, Funding acquisition, Writing – review and editing                                                                  |
| 773<br>774 | Hirmas, D.R.: Data Curation, Funding acquisition, Methodology, Investigation, Writing – review and editing                                                   |
| 775        | Johnson, K.: Data Curation, Investigation, Writing – review and editing                                                                                      |
| 776        | Kerins, D. : Investigation, Writing – review and editing                                                                                                     |
| 777        | Pachon, J: Data Curation, Investigation, Writing – review and editing                                                                                        |
| 778        | Curtis Beutler: Investigation, Writing – review and editing                                                                                                  |
| 779        | Jarecke, K.M.: Data Curation, Investigation, Writing – review and editing                                                                                    |
| 780        | Varikuti, V: Data Curation, Investigation, Writing – review and editing                                                                                      |
| 781        | Unruh, Micah,: Writing – review and editing                                                                                                                  |
| 782        | Ajami, H: Data Curation, Investigation, Funding acquisition, Methodology, Writing – review and                                                               |
| 783        | editing                                                                                                                                                      |
| 784        | Barnard, H.R.: Investigation, Funding acquisition, Writing – review and editing                                                                              |
| 785        | Flores, A.N: Funding acquisition, Writing – review and editing                                                                                               |
| 786        | Williams, K. H.: Funding acquisition, Investigation, Writing – review and editing                                                                            |
| 787        | Sullivan, P.L.: Conceptualization, Funding acquisition, Methodology, Project administration,                                                                 |
| 788        | Supervision, Visualization, Writing – original draft, Writing – review and editing                                                                           |
| 789        |                                                                                                                                                              |
| 790        | Competing Interests                                                                                                                                          |
| 791        | The authors declare that they have no conflict of interest                                                                                                   |
|            | ,                                                                                                                                                            |
| 792        |                                                                                                                                                              |
| 793        | Acknowledgements                                                                                                                                             |
| 794        | We would like to thank Reece Gregory, Nicole Hornslein, Ariel Mollhagen, and Michael                                                                         |
| 795        | Mackenzie. This material is based upon work supported by the National Science Foundation                                                                     |
| 796        | under Grants NSF 2121694 (P. L. Sullivan) and 2012796 (P. L. Sullivan); NSF 2012669 (H. R.                                                                   |
| 797        | Barnard), the Department of Energy under Grant DE-SC0020146 (L. Li; P. L. Sullivan), NSF                                                                     |
| 798        | 2121639 (S.A. Billings), and NSF 2121760 (H. Ajami; D. Hirmas). This material is partially based                                                             |
| 799        | upon work supported as part of the Watershed Function Scientific Focus Area funded by the                                                                    |
| 800        | U.S. Department of Energy, Office of Science, Office of Biological and Environmental Research                                                                |
| 801        | under Contract No. DE-AC02-05CH11231. Finally, our understanding of these sites benefited                                                                    |
| 802        | from data provided for project doi.org/10.46936/mone.proj.2023.60933/60008945 awarded to                                                                     |

- SAB and PLS by the Molecular Observation Network (MONet) at the Environmental Molecular
- Sciences Laboratory (<a href="https://ror.org/04rc0xn13">https://ror.org/04rc0xn13</a>), a DOE Office of Science user facility sponsored
- by the Biological and Environmental Research program under Contract No. DE-AC05-
- 76RL01830.

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
