# Peer review of "Soil signals of key mechanisms driving greater protection of organic carbon under aspen"

_EGUsphere, 2025_

## Author Response (AR1)

**Reviewer 1**

The presented manuscript deals with the question why there is generally more organic C (SOC) in soils under aspen than under spruce trees in North America. The strength of this work is that its methods are quite comprehensive (and described with enough details), which makes it possible to assess complex interactions. A relative weakness of the study is the limited number of sites and especially the fact that aspen and spruce sites have (naturally) different positions in the landscape, which makes it impossible to extract the "pure" effect of the tree species. The authors are perfectly honest in acknowledging this fact and the discussion is written accordingly. The main conclusions are therefore not so much based on direct proofs but rather on clusters of indicators.

The article is generally well written (for a reviewer using English as a third language). There are, however, several long, sometimes complicated sentences that would deserve a simplification (see details below).

We greatly appreciate the feedback from the reviewer and have addressed the long sentences and simplified the text as indicated below.

The bibliography list is extremely long. Could it be shortened without affecting the understandability of the text?

We removed several references including but not limited to: Kelly et al., 2001; Larson et al., 2007; Jiang et al., 2023; Neubauer et al., 2013; Tange and Johannesson 2003; Pourret et al., 2007; Zeng et al., 2021

**Details**

L. 44: more SOC under aspen than under spruce: is this per soil mass or per area? Or both?

Changed to "we observed greater SOC concentrations under Aspen", see Line 44.

L. 49: repeating "root" not necessary. Corrected

L. 66: "SOC regulates" may be misunderstood as SOC being the sole factor regulating these properties.

Changed to influences

L. 102: the effect of coarse roots on aggregates is certainly small simply because there are by far less coarse than fine roots.

This may be true; indeed, as the reviewer states coarse root abundance was dwarfed by fine roots. However, our sentence is describing known impacts of coarse vs fine roots on aggregate collapse and formation, so we would like to keep it as is. Further, we cannot know in these soils whether few coarse roots impose less of an influence on aggregates than many fine roots.

L. 142: the word "tandem" is mostly used for a two-fold combination, i.e. its use here is not wrong but a bit surprising.

Changed to "at the same time"

L. 180: there are several shapes, therefore rather "shapes represent" (plural). The whole legend of this figure is quite long.

Changed to: "Shapes represent". We appreciate that the caption is long, but due to the number of environmental factors that differ across the sites we were intentional in making the caption explicit so readers did not need to hunt through the document to understand how sites differed.

L. 193–194: it's not clear what "average yearly minimum and maximum" are. From the wording itself, it should be taking each year the minimal/maximal recorded temperature and then averaging over years. The given range is, however, very narrow for that, even more for a continental climate. Are these perhaps minimum and maximum monthly averages, then averaged over years? We simplified this sentence to: "The mean annual temperature is 0.9 °C and the mean annual precipitation is 670 mm (Carroll et al., 2018), with approximately 60% falling as snow between October and May."

L. 226: how many coring locations are there within these 100 m? Sentences was changed to: "Twice in the summer of 2022 (late June and mid-August), soil was collected from 3 auger sampling locations within ~100 meters of each pit to characterize soil chemistry".

L. 235: I would always encourage to refer to concentrations by using the word concentrations, and not indirectly by using the measurement unit like %N or %C.

One could actually also express the same parameter in e.g. g/kg. Note also that the C:N ratio is not "measured" but calculated.

We altered the text to address the comment, now L241: "We assessed SOC and SON concentrations and stocks and the likelihood of SOC degradation by microbes by analyzing bulk soil samples at 10-cm intervals. We determined SOC and SON on subsamples (~75 mg) via an elemental analyzer (Vario Macro Cube, Elementar, Ronkonkoma, NY). We used SOC and SON concentration measurements to calculate each subsample's C:N ratio. To determine stocks of SOC in each horizon, we multiplied SOC concentrations by soil bulk density obtained in each horizon. Bulk density was measured using a three-dimensional laser scanner (3D Scanner Ultra HD, NextEngine, Inc., Santa Monica, CA) following Rossi et al. (2008)."

L. 242: it would be better to specify from the beginning on that this is extractable DOC (by opposition to a DOC concentration that would be measured by lysimetry in the field).

Now line 247, changed to: "We measured extractable, dissolved organic C (DOC) to estimate organic C that can be relatively easily mobilized and transported out of the soil profiles; note that this differs from DOC measured in soil porewater using lysimeters, and instead represents a salt-extractable pool."

L. 256: prefer the SI unit Hz ( $s^{-1}$ ) to rpm. "4800 RPM" change to "80 Hz ( $s^{-1}$ )"

**L. 258 ff.: long sentence.**

Now line 265, changed to: "To assess the degree to which soil microbial communities were generating exo-enzymes that catalyze soil organic matter decay and can provide C- and N-rich compounds, we quantified potential activity rates of two such enzymes. We measured activity of  $\beta$ -glucosidase and N-acetyl- $\beta$ -D-glucosaminidase, herein referred to as BGase and NAGase, which are linked to microbial C (BGase) and N and C (NAGase) acquisition (Sinsabaugh and Moorhead, 1994; Allison et al., 2011, Stone et al., 2014), using 4-methylumbelliferyl  $\beta$ -D-glucosaminide (for NAGase) fluorescent tags. These tags were added to slurries made from approximately 1 gram of soil and pH-adjusted 50 mM sodium acetate."

L. 271 ff.: as nitrate in soils is subject to strong variations with time, please specify the vegetation and hydrology conditions at sampling time.

We added the month pits were sampled into Table 1, where the extractable nitrate concentrations are indicated to be sampled from all five pits (e.g., Aspen and Spruce).

L. 316: consider writing "oven-dry" with an hyphen. Done

L. 318: "2-4.76": an en dash should be used here instead of an hyphen (even if one could argue that this is the job of the typesetter and not of the authors...)

Thank you, corrected.

L. 331: how does ImageJ recognize roots? This is not trivial at all.

This is a manual procedure, ImageJ does not recognize the roots. We have updated the text to clarify we updated line 336: "We used ImageJ (Schneider et al., 2012) to overlay each image with a 1x1 cm grid. We then manually checked each 1x1 cm grid cell for the presence of a fine root (diameter < 1 mm) or coarse root (diameter > 1mm) and noted these presence/absence scores for each grid cell."

L. 346: use the full word "minutes" or the abbreviation "min", but not "mins". Corrected.

L. 352 ff: this calibration procedure is not clear. Transforming ppm to % should be just dividing by 10000. This works in all cases: if these ratios are for moles, for volume or for partial pressure.

Thank you for catching this, the sensors provide a millivolt reading which are converted to a %. Now Line 352 reads: "We converted O2 from millivolt reading to % by adding calibrated values to the millivolt value of O2. Each calibrated value was specific to the sensor installed and determined using atmospheric concentrations prior to installation."

L. 393: be a bit more specific here than just writing "more variable". The text should essentially be understandable also without accessing the supplementary material. Text on line 399 now reads: "Clay, silt, and sand content at the aspen sites (AS and EAG; Fig. S1) and one of the conifer sites (ESG) exhibited little variation with depth (average 33.1% clay and 18.8% sand), while the two other conifer sites had a greater sand and lower silt and clay content, particularly at depths greater than 25 cm (SS and SG; Fig. S1)."

L. 445: "similar range across depth" is not really clear to me as the ranges vary with depth according to the previous sentence, and the value given here is per area, i.e. apparently cumulated over all soil depths.

Text was updated to be more specific. Now starting on Line 452, "Across all sites, SOC concentrations ranged from 46.0-62.6 mg g $^{-1}$  near the surface (5 cm deep) to 4.8 to 29.0 mg g $^{-1}$  at depth (Fig. 4a). SOC concentrations were generally higher under aspen compared to spruce sites (p < 0.0001; Fig. 4a), but LME models also suggest that the best fit model included a significant interaction between vegetation and depth (p < 0.001), suggesting that SOC declines with depth for both vegetation types but to a greater extent under spruce compared to aspen. Stocks of SOC ranged between 0.01 and 1.31 kg m $^{-2}$  (Fig. 4b) these values did not exhibit consistent declines with depth or clear differences across cover type."

L. 451 ff: the text goes from the topic "DOC vs. SOC" to a second topic "additive vs. interaction" then back to the first topic and finally again to the second one. Consider rearranging this.

We have edited the text to be simpler, starting on Line 459: "In contrast to SOC concentrations, DOC was generally higher under the spruce stands compared to aspen. Similar to SOC, a model including a significant interaction between vegetation and depth was the best predictor of DOC values (p < 0.001), likely reflecting variable DOC values at different depths in both vegetation types (Fig. 4c). The DOC:SOC ratio also exhibited a significant interaction between vegetation type and depth (Fig. 4d; p = 0.0007). As with DOC, this significant interaction likely reflects variable ratio values for each cover type across depths."

L. 469–471: complicated sentence.

This section was update, starting on line 471: "Total soil nitrogen ranged from 0.2 mg  $g^{-1}$  at depth to 4.63 mg  $g^{-1}$  near the surface. A model including an interaction between vegetation type and depth was the best fit (p = 0.003; Fig. 5a). Aspen values were greater than those in spruce-dominated soils at all depths; the significant interaction implies that the decline with depth was greater in spruce soils.

L. 476: this range of C:N values goes from extremely low to quite high. Any comment about such a low value?

We edited the text to provide context, Line 481. "The spruce sites showed greater variation with depth with a similar mean value of 19.3 in the top 20 cm but widely variable values at the deepest points, ranging from 4.6 to 28.7 (Fig. 5c). A model including a vegetation and depth interaction was a meaningfully better fit than all simpler models (p = 0.0008), suggesting that the visibly greater variation in C:N with depth in spruce soils was a significantly different depth trend from the fairly constant aspen values. The lowest value measured, at depth in one of the spruce forests, is suggestive of soil organic matter highly-processed by microbial communities (Ziegler et al. 2017)."

Fig. 9: the increases in  $CO_2$  should be expected to relate to the decreases in  $O_2$  by a relatively constant respiratory quotient. This is not the case here. Is the respiratory quotient really so different or are there other processes, or some measurement errors?

We appreciate that you brought this to our attention, and the point here is not to discuss the apparent respiration quotient (ARQ) but instead to generally give an understanding of the range in data. This presentation of the data also does not allow us to assess the time component of how CO2 and O2 interact but rather there ranges relative to each other. Our data are suggestive of silicate weathering and the dissolution of CO2 gas into soil water, which can lead to an AQR much lower than 1 (Hodges et al., 2019). Values with similar ranges were observed in soils in Shale Hills and Garner Run (Hodges et al., 2019).

Hodges, C., Kim, H., Brantley, S.L. and Kaye, J., 2019. Soil CO2 and O2 concentrations illuminate the relative importance of weathering and respiration to seasonal soil gas fluctuations. *Soil Science Society of America Journal*, 83(4), pp.1167-1180.

L. 539: it does not make much sense for me to compare aspen with granite. Or do you mean spruce?

Good catch, changed to: "the spruce"

Fig. 10: the aggregate size classes seem not to add to 1, i.e. there is also a non-aggregated fraction. It would be at least as interesting to know the proportion of non-aggregated soil than to give the proportions of the size classes.

We respectfully disagree. It is unclear what we would learn from information about the non-aggregated portion of the soil plotted by depth. More importantly, however, the portion that falls below the bottom sieve is not completely unaggregated and we would not be able to separate the unaggregated from the aggregated portion of that material using the standard procedure for wet-stable aggregate-size distribution that we followed. The way we have represented this data is also fairly standard [see section 2.6.2.3 in the standard method described by Nimmo and Perkins (2002)].

L. 564: the word "further" is not really wrong as the verb "suggests" follows the verb "indicate" on L. 559. However, the sentence is so long that it comes quite in a surprising manner here.

We rephrased this section, changed to "By integrating knowledge from biology, pedology, hydrology, and soil chemistry we were better able to understand how multiple factors interact to drive observed SOC patterns in aspen and conifer montane forests. Our data indicate that differences in SOC protection give rise to often observed patterns of elevated SOC storage in soils under aspen compared to those in conifer stands (Woldeselassie et al., 2012, Laganiere et al., 2013, Boca et al., 2020, Román Dobarco et al., 2021). Furthermore, our study suggests that aspendominated soils may experience enhanced degrees of microbial transformation of SOC, with the products of those transformations exhibiting a greater tendency to reside in relatively small aggregates and thus protect carbon to a greater degree (Fig. 12)."

L. 608 ff.: long sentence. This is a bit counter-intuitive as larger aggregates would be expected to split into small aggregates and thus be at least as well able to protect DOC than such small aggregates. Does this deserve any comment? Simplified the sentence and added additional discussion. Changed to: "Literature hints that the larger size aggregates (Fig. 10c) and greater propensity for C to form large aggregates (Fig. 11b) observed in the spruce-dominated soils at our sites may be due to a greater abundance of particulate organic matter (POM) in spruce compared to aspen forest soils (Cotrufo et al., 2015; Cotrufo et al., 2019); this may be the case if spruce litterfall is more difficult to decompose."

L. 648: the word "confounded" would be good if this became a problem for a regression analysis. Here it is just about describing the situation and the word "related" would probably be better.

Rephrased to: "In our study aspen cover co-occurs where soil temperatures are warmer"

L. 670 ff.: long sentence.

Rephrased to: "Increases in stream water DOC concentration can harm global water quality by altering light and thermal regimes, nutrient cycling (e.g., Morris et al., 1995; Cory et al., 2015), the transport and bioavailability of heavy metals (e.g., Dupré et al., 1999; Trostle et al., 2016), and creating harmful disinfection byproducts (Leonard et al., 2022)."

L. 723: I don't understand the use of the word "import" in this context. Rephrased to: "can be important"

**Reviewer 2**

The authors present a comprehensive experimental investigation of the mechanisms driving higher SOC stocks under aspen compared to spruce land cover in the Coal Creek Watershed, Colorado, USA. They examine a range of biological (e.g., microbial biomass, enzymes, roots), chemical (e.g., pH, CEC), and physical (e.g., texture, aggregate size distribution) factors influencing SOC persistence across the two land cover types. Their findings suggest that increased microbial activity, coupled with enhanced mechanisms protecting SOC transformation products, contributes to greater SOC stocks under aspen. This study provides valuable insights into local-scale controls on SOC persistence and offers a rich dataset integrating biological, chemical, and physical factors, which can support future research, including modeling efforts. My comments are mostly minor, focusing on clarifications and suggestions for future investigations.

We greatly appreciate that you value you this work and see how it can support future research and modeling efforts.

I believe the sites are only discussed in the caption of Figure 1. I would include a brief mention of the sites and reference again to Figure 1 at the end of section 2.

Agreed, we have added a few sentences to the end of section 2 make this addition. Line 210 "We focused on five sites during this study. Three of our sites lie within the main drainage of Coal Creek including two spruce sites and one aspen (aspen sandstone (AS), spruce sandstone (SS), and SG (spruce granite). The last two sites are located in Elk Creek, a sub-catchment of Coal Creek, which include one granite and one aspen site both underlain by granite (Elk spruce granite (ESG) and Elk aspen granite (EAG). While ESG is on a dominantly south facing slope, it is north facing within the Elk Creek catchment."

The introduction frequently alternates between MAOC and SOC, sometimes implying they are interchangeable. The authors note that a large fraction of SOC is composed of MAOC in aspen (lines 74–77), but is there data on the extent to which MAOC dominates the SOC pool in aspen forests? What about in spruce?

Thank you for pointing this out. We added Line 77: "For example, Román Dobarco and Van Miegroet (2014) found no statistical differences in SOC concentrations between aspen and conifer soils but did find statistically higher concentrations of MAOC under aspen compared to conifers (66% compared to 48% MAOC to SOC, respectively)."

It is unclear how DOC loss was estimated or inferred at the two sites. Were volumetric water content (VWC) and DOC profiles used to calculate DOC leaching, or was some form of water and DOC balance applied? If leveraging VWC data, measurements of the hydraulic conductivity curve could be highly informative.

We have gone through the paper to provide greater clarity in multiple locations about our approach. By looking at salt-extractable DOC throughout soil profiles, we gain a sense of the fraction of the SOC pool that is potentially solubilizable. We thus can understand what SOC is potentially mobile via water flows throughout the profiles. We did not use VWC to estimate the amount of DOC moving through the system, but rather use [DOC] and our knowledge of conditions at the site to infer the potential for DOC loss under the two cover types. Under the conifers, we observed greater [DOC], greater sand content and thus greater Ksat at depths greater than 25 cm, and generally higher soil moisture (in part due to less evapotranspiration compared to aspen). We thus infer that the potentially mobile DOC fraction of conifer SOC pools is more able to move down through these profiles than in aspen forests.

The analysis of aggregate sizes and SOC is particularly interesting but requires further elaboration. The authors attribute greater SOC stability in aspen soils to

their smaller aggregate sizes. However, soil aggregation generally enhances SOC persistence by forming physical barriers against decomposition. Wouldn't larger aggregates offer greater protection? Additionally, in field conditions, smaller aggregates may either exist independently or be nested within larger aggregates in a hierarchical structure. How might this organization affect the role of aggregation in SOC protection?

Thank you for these questions, one thing to consider is that smaller aggregates tend to have radiocarbon-older SOC in them. So though larger aggregates might have more carbon, it does not appear to be stable at the scale of the whole profile. To address the first question, larger aggregates might offer protection to a greater amount of SOC, but they don't offer greater protection to the SOC they have. To address the second question: yes, there seems to be a hierarchical structure to some large aggregates. Finally, our separation method breaks apart non-water-stable aggregates, though, so we cannot know if smaller aggregates were residing inside larger ones prior to analysis. Thus, using our data we cannot offer a deeper understanding of how this structure may affect SOC protection.

Finally, the interpretation of Figure 11 requires caution, as the presented ratio may be misleading without information on SOC distribution across different size fractions. A useful approach for future studies would be to measure SOC within each size fraction. I believe this would provide the information and understanding of the patterns Figure 11 aims to illustrate.

We understand and have updated caption 11 to be clearer about how to interpret that figure. Also, it is true that looking at SOC within each aggregate can be helpful for understanding where SOC resides in soil aggregates. Please see Billings 2006 figure which examines how SOC is distributed among different size classes.

Line 602 "Please note each aggregate size class is divided by the total SOC, not the carbon associated with each size class."

Billings, S.A. (2006). Soil organic matter dynamics and land use change at a grassland/forest ecotone. Soil Biology & Biochemistry 38, 2934–2943.

**Reviewer 3**

The manuscript describes a study on the potential factors driving higher soil organic C (SOC) levels in soils under aspen compared to soils under spruce in North America. The strength is the truly comprehensive nature of the study combining methods and results from pedology, biogeochemistry and soil physics in a complementary fashion. A weakness is the number of sites but considering the amount of work it takes to collect such data, this is understandable, and, in my view does not diminish the quality of the work. Another weakness is the natural aspect of aspen and spruce sites but the authors do a good job in addressing this issue.

We greatly appreciate that you see the value in this work and acknowledge our efforts to underscore the limitations of the studies design.

A scientific question I have is in regard to soil temperature and moisture in aspen vs conifer stands. Aspen soils were found to be drier than conifer soils. I was curious how this could affect microbial activity. The authors addressed the issue that southern aspects on which aspen were found had a longer vegetation period but would the lower soil moisture play a role in equalizing the potential for microbial activity under both overstory types? I assume that the drier conditions were still quite favorable for microbial activity but would have liked to see this briefly addressed in the discussion.

Thank you for asking this question. During the growing season soil moisture volumetric fraction in the shallow soils was 0.10-0.25 under the Aspen and 0.15-0.30 under the spruce, while the aspen surface soil average temperatures often reach 12.5-15 C in the shallow aspen soils compared to 7.5 C under the spruce. These values reflect our focus on surface soils as these had the biggest contrasts. This almost doubling of near surface temperature with drier but still moist soils would suggest that temperature would likely play a larger role than moisture. Though the impact of lower moisture conditions could mitigate any differences in microbial decomposition rates with stand types due to temperature difference, aspen soil moisture appears sufficiently moist to support microbial activity responses to the higher temperatures.

Added Line 697 "We note that the volumetric fraction of soil moisture was also lower in the aspen, particularly at the shallowest soils, but that aspen soils appear to stay sufficiently moist (0.10-0.25 under aspen vs. 15-0.30 under the spruce) to support microbial activity responses to the higher temperatures."

Technical comments:

L115: the first author name in the reference is wrong, it should be Mikutta.

Thank you for catching that typo, change to Mikutta.

L236: did the soil have carbonates?

No, they don't have carbonates. We added this to line 205. "Soils are predominantly mapped as carbonate free Alfisols, Mollisols, and Inceptisols (Soil Survey Staff, 2023)."

L244: Maybe I'm exaggerating here but I think for the sake of reproducibility the ionic strength, electrical conductivity or concentration of the added salts should be mentioned. Or was it exactly as described by Laengdsmand? How was soil brought to field capacity, was it with the same simulated rainwater?

The rainwater extracts were comprised 47.9  $\mu$ M NaNO3, 4.69  $\mu$ M KCl, 23.81  $\mu$ M CaCl2 \* 2H2O, 12.09  $\mu$ M MgSO4 \* 7H2O , and 18.24  $\mu$ M (NH4)2SO4 and adjusted to a pH 4.2  $\pm$  0.5 using HCL. Field moist soils were subjected to extract, they were not brough back up to field capacity this has been altered in the text.

L248 and 256: What material were filters made from?

First instance, now line 258: "Samples were filtered through 0.45  $\mu$ m nylon syringe filters and 50 ml acid washed syringes."

Second instance, now line 265:" These samples were filtered through a 0.45  $\mu$ m polyethersulfone (PES) filter and their DOC concentration was determined via colorimetry (Bartlett and Ross 1988) on a Synergy HT microplate reader (Agilent, USA)."

L290: which pH method was used – H2O, KCl, CaCl2?

Now line 298: "The soil pH was determined in a 1:1 H2O soil slurry (Soil Survey Staff, 2022)."

L304: I assume the soil was crushed instead of ground?

Correct, soil was sieved, and big clods were broken up. Now Line 312 states: "Five grams of soil was sieved to 2 mm, and organic matter was removed by treating samples with 30% hydrogen peroxide"

L362: The methods and results sections would benefit from clarifying the samples size used in the models. Is N=5?

This is correct, the line 377 was modified: "We used linear mixed effects (LME) methods via the R package lme4 (Bates et al., 2014) to assess the influence of vegetation type, depth, and their interaction on soil abiotic conditions (N=5), various forms of soil nitrogen and C and  $\delta^{15}$ N, ASD, and root abundances."

L593: I don't think it's the proper reference for the statement. Also this name appears in two variants in the text, e.g., L641.

Thank you for catching this, it has been modified to Moore et al., 2006; also the name has been fixed to Boča throughout the manuscript.

---

## Author Response (AR2)

**Response to Associate Editor**

The manuscript is generally well written. However, an unnecessarily convoluted way to express things makes sometimes the reading difficult to follow. I suggest that the authors revise thoroughly their text to convey a scientifically sound clear message throughout the ms.

We greatly appreciate the detail and feedback from the AE and have made the appropriate changes. Below we respond to each of your suggestions and also include a track changes document.

L49-50 'Exo-enzyme data indicate that aspen soil microbes exhibited greater effort to seek organically-bound resources'; what do you mean by greater effort? Revise sentence and re-write.

Changed to "Exo-enzyme data indicate that aspen soil microbes likely access more organically-bound resources".

L 55 'and associated limitations on potential DOC export'. What do you mean?

Changed to "Our data suggest enhanced biotic activities in aspen-dominated forest soils that promote both chemical and physical protection of SOC in aspen- relative to spruce-dominated forests, which may have implications for DOC export."

L148 Define the concept of 'critical-zone-approach'

Changed to "Here, we use a holistic, critical-zone approach —integrating physical, chemical, and biological processes from the vegetation canopy to bedrock (Chorover et al., 2007)—drawing on data from biology, hydrology, pedology, and other disciplines to understand SOC dynamics and drivers."

L222 'affects' or 'effects'?

Changed to "effects"

L255 'rainwater' or 'extracted soil solution'?

Changed to "extracted soil solutions"

L342 substitute '1x1 cm' with '1 cm x 1 cm'

Changed to "1 cm x 1 cm".

L351 Sensor data. Consider adding them to Table 1

We added mention of table 1 to this section, the sensor data were already listed in the table on the 4th row.

L489-492 Not clear. Revise.

Changed to "Including the interaction between vegetation type and soil depth improved model fit (p = 0.0008), indicating that C:N varied more with depth in spruce soils than in aspen soils, where values stayed fairly constant. The lowest C:N value, found at depth in one of the spruce forest, suggests that the soil organic matter there has been heavily processed by microbes (Ziegler et al. 2017)."

L497 Substitute 'the aspen' with 'aspen'

Done.

L512-513 'suggesting that the greater total root abundance under aspen was driven by the coarse root fraction.' Figure 6 shows that both fine roots and coarse roots contribute to the total higher abundance of roots in aspen than spruce.

We altered the sentence to reflect that statistics. "In contrast, vegetation type offered no additional explanatory power to the depth-dependent fine root abundance (p > 0.05; Fig. 6b), suggesting that the greater total root abundance under aspen was driven more by the coarse root fraction."

L605 substitute 'thus fate' with 'thus C fate' Added

L606 'biotic action' or 'biotic processes'? Changed to "processes"

L614-615 'are typical'. What do you mean?

Changed to "Differences in litterfall composition and thus decay rates across aspen and conifer species have been widely reported, with generally lower lignin and higher nitrogen content in aspen litter (Moore et al., 2006)."

L634 'if' or 'since'?

Changed

L653 'to depth' or 'in deeper soil layers'?

Changed

L658 'unit total'? What do you mean?

Changed to "Our data revealed little direct correspondence of root abundance with SOC. However, per unit root abundance, spruce soils appear to harbor more DOC compared to aspen (Figure 7). This pattern—especially evident in total and fine root abundance—suggests that DOC moves more readily through spruce soil profiles, potentially leading to greater DOC losses to stream water compared to aspen-dominated soils."

L670 'In our study aspen cover co-occurs where soil temperatures are warmer...' it co-occurs with what?

Changed to "occurs".

L682 substitute 'down profile' with 'down the soil profile' Changed

Revise all figure legends to ensure that they are stand-alone so that they can be understood without recourse to the main text (What? Where? Why?).

Edits have been made.

Simplify Fig. 1 legend, e.g. sites on granite bedrock, blue triangles; sites on sandstone, pink circles; AS, aspen on sandstone; SS, spruce on sandstone... etc

**Figure has been updated.**

Figures 4-5-6-7-9-10-11 Mean and standard deviation are not the main information. Revise highlighting the main information. Ex. Figures 4 a) soil organic carbon (SOC) concentrations, (b) SOC stock [by horizon per pit; solid lines indicate sites underlain by granite and dotted lines aspen], (c) dissolved organic carbon (DOC), and (d) the ratio of dissolved organic carbon to soil organic carbon (DOC:SOC) with depth under two different vegetation types, aspen (orange) and spruce (green) at the Coal Creek catchment, Colorado, US. Values represent mean +/- standard deviation.

We have updated all figure captions highlight the main information.

Figure 9. What does this graph represent? It is not clear.

**We have updated the caption to be clearer**

"Soil gas concentrations of  $O_2$  (%) and  $CO_2$  (%) at aspen (orange) and confer (green) sites during the growing season at depths 15 cm (light), 45 cm (medium), and 110 cm (dark) at the Coal Creek catchment, Colorado. Values represent mean (points) +/- standard deviation

(bars) with lines connecting depths within each profile. The shallowest depth of each site is labeled: AS, Aspen Sandtone; EAG, Elk Aspen Granite; ESG, Elk Spruce Granite; SG, Spruce Granit; SS, Spruce Sandstone."

Table 1: ' $\beta$ -glucosidase and N-acetyl- $\beta$ -D- glucosaminidase'. Weren't they quantified for the auger samples, i.e., third column?

These were only done for the soils collected from the pits in July of 2022. We have updated the method text to be more explicit "Bulk soil samples were collected by depth every 10 cm for the first set of pits (2020-2021), and by horizon for the second set of pits (2022). Samples were then immediately stored in a refrigerator or freezer (DOC, microbial biomass C, exo-enzyme assays, nitrate) until they could be ground, sieved to 2 mm and analyzed. Twice in the summer of 2022 (late June and mid-August), soil was collected from 3 auger sampling locations within ~100 meters of each pit to characterize soil chemistry (i.e., SOC, DOC, pH). Soils were augured at 10 cm intervals to 110 cm (or deepest possible depth), and samples were stored in coolers with ice packs in the field and transported back to the lab and stored at 4 °C (most analyses) or frozen (SOC, DOC)."

- 1 Soil signals of key mechanisms driving greater protection of organic carbon under aspen
- 2 compared to spruce forests in a North American montane ecosystem
- 3 Authors: Lena Wang1, Sharon A. Billings2, Li Li3., Daniel R. Hirmas4, Keira Johnson1, Devon
- 4 Kerins3, Julio Pachon5, Curtis Beutler6, Karla M. Jarecke1, Vaishnavi Varikuti4, Micah Unruh2,
- 5 Hoori Ajami7, Holly Barnard8, Alejandro N. Flores9, Kenneth Williams6, Pamela L. Sullivan1

[revised manuscript text omitted]

(15, 45, 110
cm)      |              |                              |
| Root Distributions                                 | X                                    | Х            |                              |
| %C and %N                                          | Χ                                    | Χ            | Χ                            |
| Extractable nitrate concentrations                 |                                      | Χ            |                              |
| δ 15 N                                  |                                      | Χ            |                              |
| рН                                                 | Х                                    | X            | Х                            |
| Effective cation exchange capacity (ECEC)          | X                                    |              |                              |
| Texture                                            | Х                                    |              |                              |
| Wet aggregate size distribution (ASD)              |                                      | Х            |                              |
| Dissolved organic carbon (DOC)                     |                                      | Х            | Х                            |
| Microbial biomass carbon                           |                                      | Х            |                              |
| β-glucosidase and N-acetyl-β-D-
glucosaminidase |                                      | Х            |                              |

240 241 3.1 Measuring Soil Organic C and Nitrogen Dynamics 242 We assessed SOC and SON concentrations and stocks and the likelihood of SOC and SON 243 degradation by microbes by analyzing bulk soil samples at 10-cm intervals. We determined SOC 244 and SON on subsamples (~75 mg) via an elemental analyzer (Vario Macro Cube, Elementar, 245 Ronkonkoma, NY). We used SOC and SON concentration measurements to calculate each 246 subsample's C:N ratio. To determine stocks of SOC in each horizon, we multiplied SOC 247 concentrations by soil bulk density obtained in each horizon. Bulk density was measured using a 248 three-dimensional laser scanner (3D Scanner Ultra HD, NextEngine, Inc., Santa Monica, CA) 249 following Rossi et al. (2008). We measured extractable, dissolved organic C (DOC) to estimate organic C that can be 250 251 relatively easily mobilized and transported out of the soil profiles; note that this differs from 252 DOC measured in soil porewater using lysimeters, and instead represents a salt-extractable 253 pool. We analyzed soil samples at 10-cm intervals to auger refusal collected at each site during 254 the growing season. Soil samples were extracted within three months of collection date. A total 255 of 7.5 g of soil at field moisture was extracted with 30 ml of simulated rainwater (Laegdsmand et al., 1999). The extracted soil solutions were comprised of 47.9  $\mu$ M NaNO3, 4.69  $\mu$ M KCl, 256 257 23.81 μM CaCl2 x  $2H_2O$ , 12.09 μM MgSO4 x  $7H_2O$ , and 18.24 μM (NH4)2SO4 
[revised manuscript text omitted]